# Fatigue Factor Assessment and Life Prediction of Concrete Based on Bayesian Regularized BP Neural Network

**DOI:** 10.3390/ma15134491

**Published:** 2022-06-25

**Authors:** Huating Chen, Zhenyu Sun, Zefeng Zhong, Yan Huang

**Affiliations:** 1Faculty of Urban Construction, Beijing University of Technology, Beijing 100124, China; 15725701865@163.com; 2Baobida IOT Technology (Suzhou) Co., Ltd., Suzhou 200041, China; zhongzefengwy@163.com

**Keywords:** concrete tensile fatigue, backpropagation neural networks, Bayesian regularization, quantitative assessment, influencing factors, fatigue life prediction, average relative impact value

## Abstract

Concrete tensile properties usually govern the fatigue cracking of structural components such as bridge decks under repetitive loading. A fatigue life reliability analysis of commonly used ordinary cement concrete is desirable. As fatigue is affected by many interlinked factors whose effect is nonlinear, a unanimous consensus on the quantitative measurement of these factors has not yet been achieved. Benefiting from its unique self-learning ability and strong generalization capability, the Bayesian regularized backpropagation neural network (BR-BPNN) was proposed to predict concrete behavior in tensile fatigue. A total of 432 effective data points were collected from the literature, and an optimal model was determined with various combinations of network parameters. The average relative impact value (ARIV) was constructed to evaluate the correlation between fatigue life and its influencing parameters (maximum stress level Smax, stress ratio R, static strength *f*, failure probability P). ARIV results were compared with other factor assessment methods (weight equation and multiple linear regression analyses). Using BR-BPNN, S-N curves were obtained for the combinations of R = 0.1, 0.2, 0.5; *f* = 5, 6, 7 MPa; P = 5%, 50%, 95%. The tensile fatigue results under different testing conditions were finally compared for compatibility. It was concluded that Smax had the most significant negative effect on fatigue life; and the degree of influence of R, P, and *f*, which positively correlated with fatigue life, decreased successively. ARIV was confirmed as a feasible way to analyze the importance of parameters and could be recommended for future applications. It was found that the predicted logarithmic fatigue life agreed well with the test results and conventional data fitting curves, indicating the reliability of the BR-BPNN model in predicting concrete tensile fatigue behavior. These probabilistic fatigue curves could provide insights into fatigue test program design and fatigue evaluation. Since the overall correlation coefficient between the prediction and experimental results reached 0.99, the experimental results of plain concrete under flexural tension, axial tension, and splitting tension could be combined in future analyses. Besides utilizing the valuable fatigue test data available in the literature, this work provided evidence of the successful application of BR-BPNN on concrete fatigue prediction. Although a more accurate and comprehensive method was derived in the current study, caution should still be exercised when utilizing this method.

## 1. Introduction

Bridge decks, highway pavements, and railway sleepers are structural components subjected to numerous repetitions of bending load cycles during their entire service life. Fatigue failure may occur even when the loads on the structures are lower than their strength under static loading. Since concrete is primarily utilized with its compressive strength, much attention has been devoted to compressive fatigue performance. With tensile strength significantly lower than compressive strength, concrete’s tensile behavior controls the fatigue cracking of concrete structures and plays a vital role in concrete durability. Therefore, the tensile fatigue analysis of concrete is of prime importance. Fatigue life is an essential indicator of fatigue performance. As a complex multi-phase composite material, concrete exhibits significant discreteness in fatigue life [1]. Moreover, since a nonlinear mapping relationship exists between fatigue life and its influencing factors [2], fatigue life estimation has become the focus of concrete fatigue research.

The conventional method of analyzing fatigue life is based on mathematical statistics and fits experimental fatigue data to a specific regression equation. Parameters considered in the fatigue life equations initially contained only the stress level. Later, the stress ratio R [3], the failure probability P [4], and the loading frequency [5] were gradually integrated for practical applications. Despite their extensiveness and complexity, the proposed equations cannot be applied to all fatigue analyses and it is difficult to ensure accuracy, and a unanimous consensus on quantitative measurement of these factors has not been achieved [6]. On the other hand, the backpropagation neural network (BPNN) has emerged as the most widely used soft computing method, automatically approximating the training data. BPNN does not need to make assumptions about the functional form [7]; it is thus feasible to improve the applicability and prediction accuracy for multi-parameter fatigue life fitting.

Although BPNN has primarily been used to predict concrete strength [8,9,10,11] and durability properties [12,13], several studies [14,15,16,17,18,19,20,21] on fatigue life prediction of concrete did demonstrate its rationality and effectiveness. Fatigue life prediction generally fits closer to the experimental results, and the model performs statistically better than the code equations [14]. However, these studies primarily focused on concrete fatigue behavior under a compressive load, and a model suitable for concrete tensile fatigue life prediction is still lacking. Moreover, BPNN prediction is affected by the mapping relationship between input and output. For example, Mohanty et al. [22] found that a three-input parameter model’s predicted fatigue crack growth rate correlates better with test results than a two-input parameter one. A mathematical analysis of selecting and determining critical input parameters is much needed.

Because the learning process of the BPNN algorithm strongly depends on the training data, its extrapolation capability is not guaranteed if a significant difference exists between the training set and the test set [14,23]. As illustrated in Figure 1, data were divided into a training set, test set, and validation set. It is possible that the trained model approximates the test set or validation set data well but shows a poor prediction for an unknown data set B. This phenomenon is called overfitting and is a common drawback of BPNN. Thus, the standard practice for fatigue life prediction with BPNN, selecting training and validation data from the same experiments, fails to demonstrate the model’s generalization capability. For the same reason, previous studies focused on fatigue under a particular loading condition; the relationship between concrete fatigue lives under different stress states has not been thoroughly investigated.

Among the many optimization methods, the regularization technique using an additional weight attenuation term in the error function of BPNN can appropriately fix the overfitting problem [17]. Bezazi et al. combined BPNN with the Bayesian technique to predict fatigue probability life, and the results showed that Bayesian training could obtain better data fitting performance [17]. The so-constructed model is a Bayesian regularized backpropagation neural network (BR-BPNN).

In order to evaluate the significance of input parameters, researchers investigated different methods. Garson [24] and Li et al. [25] analyzed the influence weight of each input variable based on the weight matrix, which could be applied to the single and double hidden layer neural network, respectively. Lopes et al. [26] took the contribution factor as a guide to screen input variables, calculated by the sum of the weights connected to an input variable. In addition to the weight matrix, scholars also use multiple linear regression (MLR) to explain the significance of input variables, taking the significance output of MLR as the input of neural networks [27,28,29]. This method can also identify the direction of parameter influence (positive or negative correlation), achieving satisfactory results based on linear and additive correlation.

Therefore, this study aimed to validate whether a successful fatigue life analysis for concrete flexural fatigue based on existing data could also be obtained with BPNN. For the common problem of localized optimization with BPNN, the Bayesian regularization optimization [30] was constructed with MATLAB to improve the accuracy of fatigue life prediction. One novelty of this study is its use of a separate data set when verifying the model’s generalization capability. Since there is no currently available mathematical assessment of the importance of various factors related to concrete fatigue life [31], the average relative impact value (ARIV) was proposed as a quantitative index. Its use was verified through the comparison with the weight equation and MLR method. Based on the data from the literature [32,33,34,35], the BR-BPNN with satisfactory generalization capability for concrete flexural fatigue was obtained and then applied to predict flexural fatigue life curves under various parameters. Finally, to investigate whether a significant discrepancy exists between fatigue properties under different loading conditions [36], the fatigue life of plain concrete in splitting tension, axial tension, and flexure loading was analyzed through BR-BPNN.

## 2. Materials and Methods

### 2.1. Data Collection and Preprocessing

In fatigue experiments, the recordable parameters mainly include R, Smax, P, *f,* water-cement ratio (W/C), sand-cement ratio (S/C), and gravel-cement ratio (G/C). It should be noted that failure probability (P) is estimated by average rank [37]. By organizing fatigue life data in increasing order, the failure probability of any sample could be calculated as in the following expression:(1)$$P=\frac{i}{N+1}$$
where *i* denotes the failure order and *N* is the total number of data points at a particular stress level.

Table 1 presented a dataset containing 274 data points obtained from concrete flexural fatigue experiments in the literature [32,33,34,35]. The data in the literature [32,33,34] were randomly divided in a ratio of 8:2, obtaining 170 training data points and 44 test data points to guarantee the network accuracy. The remaining 60 data points in the literature [35] were reserved to verify the generalization capability further.

For mutual prediction between flexural, splitting, and axial tensile fatigue in Section 3.5 with an artificial neural network, 158 additional sets of experimental data were collected from the concrete tensile fatigue literature [38,39,40,41,42,43], as shown in Table 2. Due to the approximation of stress states from actual structures and the difficulties associated with concentric loading, splitting/axial tensile fatigue test data were less abundant than their bending counterpart.

In order to improve the convergence rate of the network and avoid the deviation adjustment of weight caused by dimensional differences [44], normalization was used for data preprocessing. The scale transformation of original data was conducted according to Equation (2), which is realized by [*y*, ps] = mapminmax (*x*, $y_{\min}$, $y_{\max}$) in MATLAB, where ps represents the mapping relationship; x and y are the data before and after normalization; $y_{\max}$, $y_{\min}$ is the maximum and minimum of normalized boundary, respectively; and $x_{\max}$, $x_{\min}$ is the maximum and minimum of input before normalization, respectively.
(2)$$y=\frac{\left(y_{\max}-y_{\min})\times (x-x_{\min}\right)}{\left(x_{\max}-x_{\min}\right)}+y_{\min}$$

The output needs to be returned to the original order of magnitude for the practical application of the predicted results. Therefore, anti-normalization was performed according to Equation (3), which is realized by *x* = mapminmax (‘reverse’, *y*, ps) in MATLAB.
(3)$$x=\frac{\left(y-y_{\min})\times (x_{\max}-x_{\min}\right)}{\left(y_{\max}-y_{\min}\right)}+x_{\min}$$

### 2.2. Basic Principle of Backpropagation Neural Network

BPNN, a multi-layer feedforward network trained by the error backpropagation algorithm, can classify arbitrarily complex patterns and map nonlinear multi-dimensional functions [45]. Its essence is to calculate the error through the forward transmission of function signals among neurons and then backpropagate the error signal to correct the weight and bias of the network, repeatedly iterating until the loss function is minimized.

The topological structure of the L-layer BPNN is shown in Figure 2. The network includes the input layer (layer 1), the hidden layers (layer 2~L − 1), and the output layer (layer L), with each layer consisting of several neurons [46]. $\vec{X}=[x_{1},x_{2},\ldots x_{i}]$ represents input vector and $\vec{O}=[o_{1},o_{2},\ldots o_{k}]$ represents output vector. $w_{ij}^{\left(l\right)}$ is the connection weight between the *i*^th^ neuron in layer *l* − 1 and the *j*^th^ neuron in layer *l*; $b_{j}^{\left(l\right)}$ is the bias of the *j*^th^ neuron in layer *l*, and $\varphi^{\left(l\right)}(\cdot )$ represents the activation function of layer *l* neurons. The input and output of neurons in hidden layer *l* are $V^{\left(l\right)}=[v_{1}^{\left(l\right)},v_{2}^{\left(l\right)},\ldots v_{j}^{\left(l\right)}]$ and $H^{\left(l\right)}=[h_{1}^{\left(l\right)},h_{2}^{\left(l\right)},\ldots h_{j}^{\left(l\right)}]$, respectively. $v_{j}^{\left(l\right)}=\sum_{i=1}^{I}w_{ij}^{\left(l\right)}h_{i}^{\left(l-1\right)}+b_{j}^{\left(l\right)}$, where I is the number of neurons in layer *l*-1 and $h_{j}^{l}=\varphi^{l}(v_{j}^{\left(l\right)})$.

Suppose that the target output of the *n*-th training sample is $\vec{T}=[t_{1n},t_{2n},\ldots t_{kn}]$, the loss function is measured by mean square error (MSE): $E_{D}=\frac{1}{2N}\sum_{n=1}^{N}\sum_{k=1}^{K}{\left(t_{kn}-o_{kn}\right)}^{2}$. The weight and bias are revised through the backpropagation of errors until the loss function is minimized using $w_{ij}^{\left(l\right)}=w_{ij}^{\left(l\right)}-\eta \frac{\partial E_{D}}{\partial w_{ij}^{\left(l\right)}}$ and $b_{i}^{\left(l\right)}=b_{i}^{\left(l\right)}-\eta \frac{\partial E_{D}}{\partial b_{i}^{\left(l\right)}}$, where η is the learning rate of backpropagation [47].

The classical BPNN with a solid learning ability can achieve arbitrary fitting accuracy within the training samples [48]. However, the generalization capability of the network depends considerably on the number of neurons in the hidden layers. When the number of training samples is limited, and the number of hidden layer neurons is vast, the model could produce a poor mapping effect on non-training samples. This phenomenon is called “overfitting”; the model exhibits high training accuracy but low prediction accuracy [49]. Therefore, achieving a good generalization capability of the BPNN is an essential research issue. A theoretical formula to determine the optimal number of hidden layer neurons has not yet been proposed. By trial and error, this paper found the optimal neuron number from the range of values suggested by an empirical equation, as represented in Equation (4):(4)$$m=\sqrt{i+k}+\lambda$$
where *i* is the number of neurons in the input layer; *k* is the number of neurons in the output layer; and λ is an integer between 1 and 10.

### 2.3. Bayesian Regularization

A network with a smaller weight or bias can obtain a smoother response, which is utilized by regularization to improve the generalization capability of the BPNN effectively [50], adding a penalty term to the target problem to limit its complexity. The L2 penalty term related to the weight is generally added to the loss function in the form of ridge regression [17].

The improved loss function is shown in Equation (5):(5)$$E(w)=\beta E_{D}+\alpha E_{w}=\frac{\beta }{2N}\sum_{n=1}^{N}\sum_{k=1}^{K}{\left(t_{kn}-o_{kn}\right)}^{2}+\frac{\alpha }{2M}\sum_{m=1}^{M}w^{2}$$
where α and β are the regularization parameters; $E_{w}$ is the penalty term of the loss function; and *M* is the number of connection weights.

Weight is modified as indicated in Equation (6). When updating the gradient to realize weight decay, the weight is multiplied by a constant coefficient of less than 1.
(6)$$w_{ij}^{\left(l\right)}=w_{ij}^{\left(l\right)}-\eta \frac{\partial E(w)}{\partial w_{ij}^{\left(l\right)}}=(1-\eta \alpha )w_{ij}^{\left(l\right)}-\eta \beta \frac{\partial E_{D}}{\partial w_{ij}^{\left(l\right)}}$$

α, β dramatically affects the distribution of weight and bias. When β>>α, the decrease in the training error cannot guarantee the generalization capability, as in the classical BPNN. Conversely, when β<<α, the attenuation of the network scale and the smoother output is prone to “underfitting”. This paper used the Bayesian computing framework to adaptively modify α and β, which reduced the training error and optimized the network structure. The Bayesian technique can be expressed as in Equation (7):(7)$$\mathrm{Posterior}\;\mathrm{Probability}=\frac{\mathrm{Likelihood}\;\mathrm{Function}\times \mathrm{Prior}\;\mathrm{Probability}}{\mathrm{Normalization}\;\mathrm{Factor}}$$

It assumes that the likelihood function and priori probability satisfy the Gaussian distribution. Based on maximizing posterior probability, the L-M algorithm was used to solve the minimum of loss function $E(w_{opt})$. Then, the regularization parameters were further adjusted with the same idea using the Hessian matrix H at the minimum point $w_{opt}$ to re-verify the accuracy of the loss function [17].

The following equations were obtained with the Bayesian technique [50], where γ is the number of valid parameters that reduce loss function in the network.
(8)$$\alpha_{opt}=\frac{\gamma }{2E_{w}(w_{opt})},\;\beta_{opt}=\frac{N-\gamma }{2E_{D}(w_{opt})}$$
(9)$$\gamma =M-2\alpha tr(H^{-1}(w_{opt}))$$
(10)$$H=\beta \nabla^{2}E_{D}(w_{opt})+\alpha \nabla^{2}E_{w}(w_{opt})$$

The learning steps of BR-BPNN are shown in Figure 3, which are realized by the training function trainbr in MATLAB.

### 2.4. Average Relative Impact Value (ARIV)

The correlation between output and input significantly affects the training quality of the network [26], while a coupling effect might exist among the parameters shown in Table 1. In order to determine the variables with significant effects and avoid model complexity induced by noise parameters, ARIV was used to determine the influence of input on output, whose symbol represents the relevant direction and the absolute value represents the relative importance.

Firstly, an original input variable, including *i* samples, is increased or reduced by 10% to derive two new datasets $X_{1}=\{X_{11},X_{12},\ldots X_{1i}\}$ and $X_{2}=\{X_{21},X_{22},\ldots X_{2i}\}$, where $X_{1i}$ and $X_{2i}$ represent the dataset formed by a 10% increase or decrease in the *i*^th^ sample in this variable, respectively. Then, the BR-BPNN trained by the original training sample predicts the other two datasets, and the corresponding prediction results are $O_{1}=\{O_{11},O_{12},\ldots O_{1i}\}$ and $O_{2}=\{O_{21},O_{22},\ldots O_{2i}\}$. The ratio of the difference between $O_{1}$ and $O_{2}$ to the original output ***O*** is defined as the relative impact value matrix (**RIVM**). RIVM is finally averaged according to the number of samples to obtain the ARIV of each input parameter, as shown in Equations (11) and (12).
(11)$$RIVM=\frac{O_{1}-O_{2}}{O}$$
(12)ARIV=avg(RIVM)

### 2.5. Weight Equation

The weight matrix of a neural network can evaluate the relative importance of each input variable to the output variable [51]. For a three-layer BPNN which can complete the infinite approximation to any continuous function (in the closed interval) [52] and realize the function mapping from any input to output, Garson proposed an equation to calculate the influence weight of each input variable based on the weight matrix [24]. This paper also utilized the Garson’s connection weight division equation [24] to evaluate the influence weight of input variables, as shown in Equation (13):(13)$$I_{jn}=\frac{\sum_{m=1}^{N_{h}}\left((\left|w_{jm}^{ih}\right|/\sum_{k=1}^{N_{i}}\left|w_{km}^{ih}\right|)\times \left|w_{mn}^{ho}\right|\right)}{\sum_{k=1}^{N_{i}}\left\{\sum_{m=1}^{N_{h}}\left((\left|w_{jm}^{ih}\right|/\sum_{k=1}^{N_{i}}\left|w_{km}^{ih}\right|)\times \left|w_{mn}^{ho}\right|\right)\right\}}$$
where $I_{jn}$ is the influence weight of the *j*th input parameter on the *n*th output neuron. $N_{i}$, $N_{h}$ is the number of input and hidden layer neurons, respectively; $w^{ih}$, $w^{ho}$ is the connection weights from the input layer to the hidden layer and from the hidden layer to the output layer, respectively.

### 2.6. Multiple Linear Regression (MLR)

Multiple linear regression studies the explanatory relationship between independent and dependent variables reflected with regression equations [52]. Suppose that the random variable (*y*) changes with independent variables (*x_k_*), and that the linear regression relationship is represented by Equation (14).
(14)$$y=\beta_{0}+\beta_{1}x_{1}+\beta_{2}x_{2}+\ldots +\beta_{k}x_{k}+\varepsilon$$
where $\beta_{0}$ represents regression constant; $\beta_{1}-\beta_{k}$ are the partial regression coefficients, representing the influence degree of independent variable on dependent variable; and ε is the standard estimation error.

After constructing the regression equation, its goodness of fit and statistical significance must be verified. Some good-fit indicators of the linear relationship of regression equation are as follows: the adjusted coefficient of determination (Adjusted R^2^) which evaluates how much the independent variables can explain the variation of the dependent variable; the standard error of the estimates (SEE) which indicates the degree of relative deviation between the actual value and the estimated value; F change, which is the constructed statistic to test the significance of the whole regression model; and Sig. F change, which is the probability corresponding to the F statistic.

The overall significance of the regression model does not guarantee the same significance for each independent variable. In order to extract significant variables to optimize the model structure, further significance testing of the regression coefficient is required. The partial regression coefficient (B) and its standard errors (Std.error) indicate the effect of independent variables on dependent variables. The magnitude of B represents the degree of influence, and the sign of B represents the direction of correlation. In addition, to judge the contribution of independent variables to the variation of dependent variables, the test *T* was used to test whether the probability corresponding to the t-statistic was equal to 0, where the t-statistic was constructed as $t=\frac{B}{\mathrm{Std}.\mathrm{error}}$. The tolerance (Tol) and variance inflation factor (VIF) are indexes of multicollinearity diagnosis, and it is generally believed that linear correlations probably exist between independent variables when *Tol* < 0.1 or *VIF* > 10.

## 3. Results and Discussion

### 3.1. Selection of Hyperparameter and Function

The learning rate affects the step adjustment of the loss function [53]. A smaller value will lead to a slower convergence rate and a more extended network response time. However, if it is too large, the neural network will converge too quickly, and it could easily miss the globally optimal solution [53]. In this study, the learning rate was set to 0.01. Since the error tolerance parameter limits the neural network’s weights to avoid falling into a local optimum during the training process, the allowable error was set to 1 × 10^−6^. The maximum number of iterations was set to 10,000, considering the network’s excellent computing power. We used the default values in the MATLAB toolkit for the rest of the parameters.

The prediction effect of BR-BPNN depends on the selection of various functions. The activation function that must be continuously differentiable plays a nonlinear transformation role in the input and output [54], transforming the input of an infinite field into an output within a specified range. The learning function returns the correction value of weight and bias in each layer, considering the minimization of local errors. The training function realizes the output of the training records and calls on the learning function during the training process to correct the connection weight and bias [55]. The training is terminated when the number of iterations or the calculation error of the loss function satisfies the preset value, considering the minimization of the global error.

This paper exhaustively conducted a combined trial of various functions in the algorithm. The optimal combination of activation function and learning function was selected based on the minimum MSE of the training result. The influence of function combinations on the results is shown in Figure 4.

Notes: 1~10 represents the combination number. The legend describes the combination mode according to the activation function (the first for hidden layer, the second for output layer) + learning function. TANSIG is the hyperbolic tangent function; LOGSIG is the Sigmoid function; POSLIN is the positive linear function; and PURELIN is the linear transfer function. LEARNGDM is the gradient descent momentum learning function; and LEARNGD is the gradient descent learning function.

As seen from Figure 4, the No. 5 network structure (LOGSIG + TANSIG + LEARNGDM) had the lowest error performance with an MSE of 0.00024 and a maximum relative error of 7.89%. The final function combination is shown in Table 3.

### 3.2. Determination of the Neurons

The fatigue life N was used in the output layer to reflect the fatigue performance of concrete.

In order to obtain the influence effect of each variable, this paper analyzed the changes of the ARIV after ten rounds of random network training, as shown in Figure 5.

It can be seen from Figure 5 that the correlations reflected by ARIV of the R, Smax, P, and *f* were relatively consistent. While R, P, and *f* were proportional to fatigue life, Smax was inversely proportional to fatigue life. The most considerable absolute value of ARIV indicated that Smax had the most significant effect. The randomness of the initial weight and bias resulted in different prediction effects after multiple network training [25], so the ARIV was not constant. The ARIV of W/C, S/C, and G/C failed to show a consistent correlation, thus regarded as noise parameters. Therefore, the input layer neurons comprised the following four parameters: Smax, R, P, and *f*.

Ten network structures were constructed for the range of hidden layer neurons determined by Equation (4). The correlation coefficients of network training and generalization under these ten network structures are shown in Figure 6. The final BR-BPNN then predicted the fatigue life for tests conducted in the literature [35], and the generalization effect is shown in Figure 7.

It was found, as shown in Figure 6, that the correlation coefficient of the dataset for network training was 0.9959 (left axis) and that for generalization it was 0.9931 (right axis) when the number of hidden layer neurons was nine, both of which showed a strong correlation. This paper thus constructed the final BR-BPNN with nine hidden neurons, which demonstrated high correlation coefficients in both the training and generalization stages.

Figure 7 shows that the prediction coincided with the target curve, the MSE between the target value and the predicted value was 0.0482, and the correlation coefficient R was 0.9931. Therefore, the BR-BPNN model provided a reliable generalization capability for the following direct applications to S-N curves prediction.

### 3.3. Feasibility Analysis of ARIV

The correlation between the input parameters and fatigue life was analyzed in Section 3.2 for random training networks. To reflect the influence degree of input parameters in more depth, the final BR-BPNN was used to re-analyze the ARIV of Smax, R, P, and *f*, and the results are shown in Table 4.

It can be seen from Table 4 that the correlation between various input parameters and fatigue life was consistent with previous conclusions. The maximum stress level affected the fatigue life negatively; and the stress ratio, failure probability, and static strength were positively correlated with fatigue life. The absolute value of ARIV shows that Smax had the most significant effect on fatigue life, followed by R, P, and *f*. In this section, the ARIV results are further verified by the SPSS regression analysis and weight equation to accurately reveal the importance rank of input variables.

#### 3.3.1. Verification by Weight Equation

Table 5 lists the value of 45 connection weights of the final BR-BPNN. $w^{ih}$ comprises thirty six connection weights from four input parameters to nine neurons in the hidden layer, and $w^{ho}$ comprises nine weights from the nine hidden layer neurons to the one output parameter of fatigue life.

The influence weights of R, Smax, P, and *f* calculated from Equation (13) are 0.265, 0.501, 0.128 and 0.106, respectively. Therefore, it was concluded that the maximum stress level has the greatest influence on fatigue life, followed by stress ratio, failure probability, and static strength, which is consistent with the results from the ARIV analysis.

#### 3.3.2. Verification by SPSS Regression Analysis

Based on the correlation conclusions from Section 3.3.1 and the MLR equation in Section 2.6, three regression models were built, and the analysis results are shown in Table 6 and Figure 8. The difference between these models was primarily about whether to include the mixed design parameters as independent variables.

Notes: Multicollinearity may exist between various input variables. In case of serious multicollinearity, SPSS automatically excludes a variable with a tolerance of much lower than 0.1 to avoid affecting the significance analysis. This variable is indicated as the “Removed Variable”.

It can be seen from Table 6 that Models 1–3 show a significance of less than 0.05, indicating that at least one factor of the regression model has a significant impact on fatigue life, which is statistically significant. Model 1, considering all seven parameters, has a desirable fitting effect with an adjusted coefficient of determination of 0.937, showing that Smax, R, P, *f*, S/C, and G/C can explain 93.7% of the variation in fatigue life. Notably, W/C was removed automatically in the regression analysis due to a low tolerance value (Tol=0.000<<0.1), which indicated that W/C is not suitable as an explanatory variable for fatigue life due to severe multicollinearity between W/C and other variables. Ignoring mixture ratio variables in Model 2, the coefficient of determination only reduced by 0.6%, compared with Model 1, and SSE only increased by 4%, further demonstrating that the mixture ratio has little effect on fatigue life. Model 3 incorporated mixture ratio parameters but ignored P and *f*, presenting a more considerable variation. The adjusted R^2^ reduced by 6.6% compared to Model 1, verifying that P and *f* can better explain fatigue life than the mixture ratio. The goodness of fit for Model 4 was poor, and W/C, S/C, and G/C could only account for a 4.7% variation in fatigue life. It can be concluded that the mixture ratio was not suggested as an independent variable in the fatigue life analysis model.

Figure 8 shows the T-test results of partial regression coefficients. As shown in Figure 8a, the significance value for all input parameters was less than 0.05, indicating that each of the four parameters (Smax, R, P, *f*) significantly affect fatigue life. In Figure 8b, B’s positive and negative display indicates that Smax has a negative impact, while R, P, and *f* have a positive impact. Since B_Smax_ = −0.873, B_R_ = 0.262, B_P_ = 0.102, B*_f_* = 0.067, the degree of influence of each parameter is: Smax > R > P > *f*. These results are also consistent with those from the ARIV and weight equation analyses.

#### 3.3.3. Comparison between Various Methods

The ARIV analysis, weight equation analysis, and SPSS regression analysis show a consistent observation, which verifies the correctness of the final BR-BPNN. It also confirms the feasibility of ARIV in determining the correlation between inputs and outputs and the relative importance of various input parameters affecting fatigue life. However, whether ARIV is more practical than the other two methods requires a further exploration of the advantages and disadvantages of the three methods in a multi-parameter significance analysis.

The weight equation analysis evaluates the relative importance of the input parameters, which thoroughly explains the structural significance of the weight matrix in the neural network. As intuitive and concise as the method is, Equation (13) is only valid for the three-layer neural network, and the equation’s operation with absolute values cannot show a positive or negative correlation between input parameters and fatigue life.

The mathematical meaning of the SPSS MLR analysis is clear, and the influence of input parameters on fatigue life can be evaluated from the aspects of correlation and importance. However, this method assumes a linear fit of the regression equation without considering other types of relationships. It cannot reflect the actual mapping relationship well when the noise factor is included, which can be considered a limitation [56], as shown in Table 7. The collinearity statistics of *f* and G/C in Model 1 show that the model demonstrated multicollinearity. As a result, the significance test of the independent variables and the model prediction function become meaningless.

Although the ARIV analysis of a neural network is strongly model-dependent, it can deal with nonlinear relationships and could preliminarily screen out the noise variables with insignificant effects through trial calculation. It also helps to construct the optimal network, a feedback mechanism to evaluate the correlation between the input and output parameters and the relative importance of various input parameters. Therefore, compared with the weight equation analysis and SPSS regression analysis, the ARIV analysis has a broader scope of application, and it is recommended for future variable effects analysis.

### 3.4. S-N Curves Predicted by BR-BPNN

Based on experimental results, the conventional research on flexural fatigue of concrete aims to fit maximum stress level versus fatigue life equation according to a failure probability of about 50%. The experimental data and S-N curves from various studies are shown in Figure 9.

Because fatigue tests are generally time-consuming and expensive, the number of specimens in individual studies is limited, as shown in Figure 9. Therefore, it is desirable to combine them to obtain a more accurate concrete fatigue life prediction under various stress states and guarantee rates.

This paper utilized the generalization capability of BR-BPNN to predict concrete fatigue life in different states by providing different values of Smax, R, P, and *f*. In theory, the possible combinations of parameters are infinite. For practical demonstration, considering the input data characteristics for network training, the following text only shows the generalization effect when R = 0.1, 0.2, 0.5; *f* = 5, 6, 7 MPa; P = 5%, 50%, 95% for concrete flexural fatigue.

#### 3.4.1. S-N Curve under 50% Guarantee Rate

Figure 10 shows the S-N curves predicted using the BR-BPNN under the 50% guarantee rate for various combinations of *f* and R. Besides the generalization curves, experimental data points from the literature are also shown in the figure, and different elements in the literature are distinguished by shape. At a specified *f*, to facilitate a comparison between the generalization curve and the experimental data, both the data points and curve under the same R were assigned a unified color, black for R = 0.1, blue for R = 0.2, and red for R = 0.5.

In accordance with the conclusions of Section 3.3, Figure 10 details the relationship between fatigue life and its affecting factors. It can be seen that the S-N curves generally agree well with the corresponding reference data. As S decreases, the fatigue life increases, although the relationship is not linear. This pattern is consistent with experimental observations [32,33,34,35]. Moreover, the fatigue life increased with an increase in R and *f*. Because few data points were available for R = 0.5 in the network training dataset, the generalization effect of the S-N curve with R = 0.5 was less satisfactory than that of R = 0.1.

#### 3.4.2. Probability Distribution of Fatigue Life

By changing P in BR-BPNN, a reliability analysis could be conducted to evaluate the failure probability of concrete specimens under fatigue load. The S-N curve under the P of 5%, 50%, and 95% generalized by BR-BPNN is shown in Figure 11. Similarly, the prediction curves under the same P were assigned a unified color, black for P = 50%, blue for P = 5%, and red for P = 95%.

This section discusses the probability distribution of concrete flexural fatigue life. It can be seen from Figure 11 that the P of 5% curve represents a lower limit of fatigue life and the P of 95% curve is an upper limit of the fatigue life. The generalization capability of the network for the fatigue life probability analysis is optimal for *f* = 5 MPa. When *f* = 6 MPa, the fatigue life probability analysis at R = 0.1 and 0.5 can be achieved. For *f* = 7 MPa, a satisfactory analysis can be obtained only at R = 0.1.

As seen from the S-N curves, the predicted results are consistent with corresponding reference data [32,33,34,35] when a strong correlation exists, which provides a comprehensive method with which to establish the probabilistic fatigue life curves from existing data.

### 3.5. Mutual Prediction of Flexural and Tensile Fatigue

Zhao et al. concluded that there were no significant discrepancies in fatigue properties for high-strength concrete under splitting tension, axial tension, and flexure [36]. To verify whether the conclusion also applies to regular grade concrete, BR-BPNN was used to mutually predict the flexure, splitting tension, and axial tension of fatigue life.

It should be noted that the axial/splitting tensile strength of concrete was less than the flexural strength. Since the model is only valid within the range of the input parameters, extrapolation beyond these ranges could not be performed [14]. Suppose the static strength is considered an input parameter while training the network. In that case, it is foreseeable that the generalization effect of the network cannot be guaranteed due to the significant difference between the tensile strengths of different stress states, as shown in Figure 12. The predicted fatigue life from the model trained from flexural fatigue data is generally lower than the target value of axial tension data when *f* is considered an input parameter. The correlation coefficient between prediction and target is only 0.779, which shows a poor simulation effect.

This section thus utilizes the three parameters of Smax, R, and P as inputs to construct a revised BR-BPNN. Due to the reduction in the input parameters, the number of hidden neurons can be appropriately increased to improve the network complexity and ensure training accuracy. When 20 neurons in the hidden layer were selected throughout the trial, the revised BR-BPNN provided a fair prediction effect, as shown in Figure 13.

Figure 13a shows that the model based on flexural fatigue data approximates satisfactorily the fatigue life of the splitting state in the range of Smax < 0.9, and the correlation coefficient can reach 0.915. Similarly, good feedback was obtained only in part in the data range for fatigue life prediction under axial tension. This range covers data points with Smin of 0.1 and 0.15, and Smax of 0.6~0.85, with a correlation coefficient of 0.944.

Figure 13b shows that the network trained based on axial/splitting tensile fatigue data has a good prediction effect on the flexural fatigue life. The overall correlation coefficient between the prediction and experimental results is 0.917.

It is noted that Smax for the majority of axial/splitting tensile fatigue data lay within 0.65~0.90, while those in Reference [43] had a Smax of 0.3~0.6. For flexural fatigue, Smax varied from 0.6~0.9. This difference could explain why axial/splitting tensile fatigue data could account for flexural fatigue life well but not vice versa. An analysis was re-performed with those data from Reference [43] excluded, and the results are shown in Figure 14.

Figure 14 demonstrates that the model based on flexural fatigue data approximated the fatigue life of the axial/splitting states, and the network trained based on axial/splitting tensile fatigue data led to a good prediction effect on the flexural fatigue life. The overall correlation coefficient between the prediction and experimental results are 0.996 and 0.985 for Figure 14a,b, respectively.

In this section, flexural fatigue data are presented under certain circumstances. It verifies that a similar trend exists for fatigue lives between different tensile stress states; thus, to some extent, the experimental results of splitting tension, axial tension, and flexure could be combined in a future analysis. This observation is meaningful for the prospect of expanding the test dataset and promoting an understanding of a unified failure mechanism of concrete tensile fatigue.

### 3.6. Limitations and Future Work

As discussed in Section 3.5, it was observed that satisfactory prediction results cannot be guaranteed unless the data to be predicted are within the scope of the training data. This finding is consistent with the caution recommended in previous research [14,23], reflecting that the artificial neural network dramatically depends on the quantity and quality of training data. Therefore, it is crucial to collect data with broad coverage, and the application of BPNN models should be exercised with caution, being limited to only the range of input parameters that have been trained.

This study provided a comprehensive method with which to establish the probabilistic fatigue life curves from existing data. The work conducted only involved BPNN with Bayesian regularization. In addition to BR-BPNN, other methods such as particle swarm optimization and genetic algorithm are available to facilitate neuron networks to reach global optimization [57]. Furthermore, abundant soft computing methods have been successfully applied to various materials. These advanced techniques include regression tree, random forest, support vector machine, extreme learning machine, genetic programming, and Gaussian process regression [2,45,46,58]. With the help of deep learning, more work is foreseeable in the future to broaden the application under different experimental conditions, to concrete with different material compositions, and eventually to structural components [20,54,59].

Research on the combination of ANN and traditional statistical analysis is also warranted. An artificial neural network can effectively extract the information contained in the data and describe the nonlinear relationship between characteristic parameters and fatigue life. However, a “black box” does not reveal much about the underlying mechanism for most practical applications [18]. Traditional statistical methods can derive the analytical probability distribution of fatigue life with physical meaning, representing fatigue life with normal distribution or Weibull distribution. Therefore, in the follow-up study, it is necessary to integrate the advantages of these two methods.

## 4. Summary and Conclusions

The tensile fatigue characteristics of concrete materials represent fundamental data for designing, assessing, and retrofitting concrete structures such as bridge decks vulnerable to fatigue cracking under repetitive loading. As fatigue life is affected by loading and material properties that are nonlinearly interwoven, the conventional statistical method shows a limited capacity to consider all these factors accurately. BPNN, which does not require assumptions about the mathematical functions, on the other hand, has become a promising alternative for fatigue life prediction in recent years. However, successful applications of BPNN on concrete flexural fatigue are still preliminary, and a comprehensive method of selecting and evaluating various input parameters is desirable. One of the most common shortcomings of BPNN is termed “overfitting”, where the model shows excellent accuracy in training but predicts poorly for an unknown data set. With an additional weight attenuation term in the error function, the Bayesian regularization technique has been shown to solve the overfitting problem. Garson and other researchers proposed various methods, including MLR, to assess the significance of input parameters quantitatively.

This study utilized BR-BPNN to build a viable model, provided three methods to evaluate parameters affecting flexural fatigue life, and predicted fatigue life under different experimental conditions. Our analysis was based on an input dataset with 432 datapoints obtained from experiments reported in the literature. Background knowledge on backpropagation neural networks, Bayesian regularization, average relative impact value, weight equations, and multiple linear regression was briefly introduced. After properly selecting hyperparameters such as activation, learning, and performance functions, we constructed the optimal model. The optimal BR-BPNN comprises one input layer of four parameters, one hidden layer with nine neurons, and fatigue life as the only output. The correlation coefficient in the stage of network training can reach 0.9959. When verifying the generalization capability of the network, it demonstrated a strong correlation of 0.9931, indicating that the network has high accuracy in both the training and generalization stages.

We used ARIV to evaluate the relative importance of parameters. In analyzing multiple variables directly affecting fatigue life, Smax was found to have the most significant impact on concrete fatigue life among the seven parameters considered, followed by R, P, and *f*. The mix ratio is regarded as the noise parameter. These findings are also verified by Garson’s weight equation analysis and SPSS MLR analysis.

The constructed BR-BPNN model is capable of predicting the concrete flexural fatigue life for any acceptable level of failure probability. Generally, the predicted S-N curves corresponding to 50% failure probability agree well with the reference data and empirical equations. The S-N curves under the 5% and 95% failure probability could be practically taken as the lower and upper bound limits of concrete flexural fatigue life, which provides a reference range of expectations when evaluating the fatigue performance of concrete subjected to fatigue loading.

This study also verifies the compatibility of tensile fatigue tests on plain concrete under different stress states. The prediction of axial/splitting tensile fatigue life from a neural network based on a flexural tensile fatigue dataset shows applicability for splitting tensile fatigue data with Smax < 0.9 and axial tensile fatigue data with Smin of 0.1 and 0.15 and Smax between 0.6 and 0.85. On the contrary, the overall correlation coefficient between predicted results and experimental values reached 0.917 when predicting flexural fatigue life from the other two modes of the tensile fatigue dataset. When those data falling out of the input range are excluded from analysis, flexural fatigue data and axial/splitting tensile fatigue data agree very well, with an overall correlation coefficient of up to 0.99.

It was noticed that the most suitable BR-BPNN for different research problems might not be identical. When predicting concrete flexural tensile fatigue, the model with four input parameters (Smax, R, P, and *f*), one output parameter (N), and one hidden layer with nine neurons had the best performance. However, in the mutual prediction between flexural tensile fatigue data and axial/splitting tensile fatigue data, the optimal BR-BPNN consisted of three input parameters (Smax, R, and P) and one hidden layer with 20 neurons.

By utilizing the valuable fatigue test data available in the literature, this paper demonstrated that BR-BPNN could successfully predict the tensile fatigue life of concrete under various conditions. For practical applications of fatigue test program design and fatigue life assessment, BR-BPNN could replace part of the expensive and time-consuming fatigue experiment and play the role of a supplementary test. In addition to the inherent defects of the artificial neural network as a “black box”, BR-BPNN has high requirements in terms of the number of samples required and the quality of training data. When sufficient sample data becomes available, more material parameters, more complex experimental conditions, more types of concrete materials, more sophisticated soft computing techniques, and perhaps mechanics-based models could be incorporated for future research.

## Figures and Tables

**Figure 1 materials-15-04491-f001:**
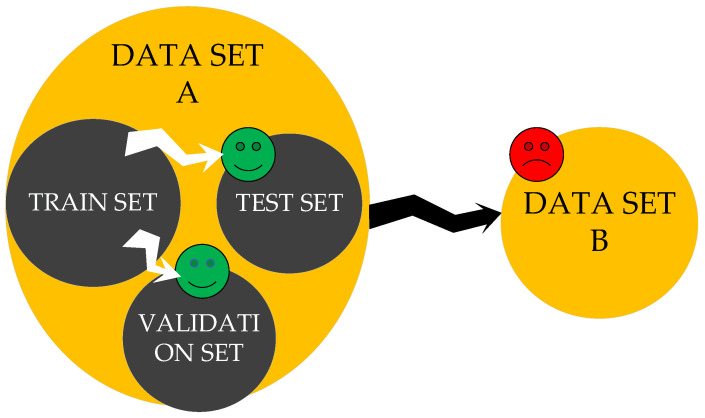
Sketch of poor extrapolation of the predictive model.

**Figure 2 materials-15-04491-f002:**
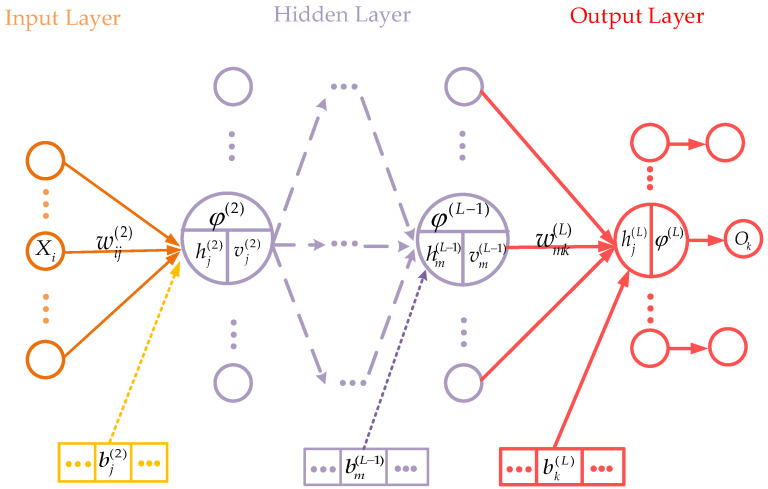
Topological structure of BPNN.

**Figure 3 materials-15-04491-f003:**
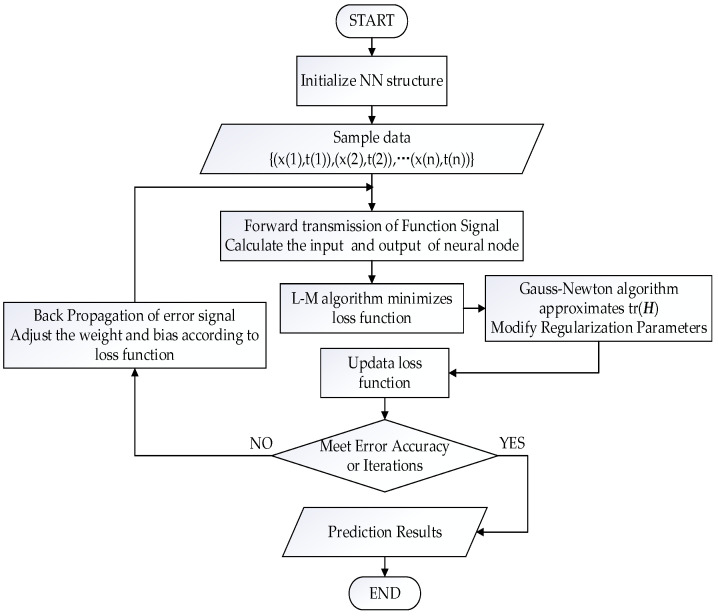
Learning flow chart of BR-BPNN.

**Figure 4 materials-15-04491-f004:**
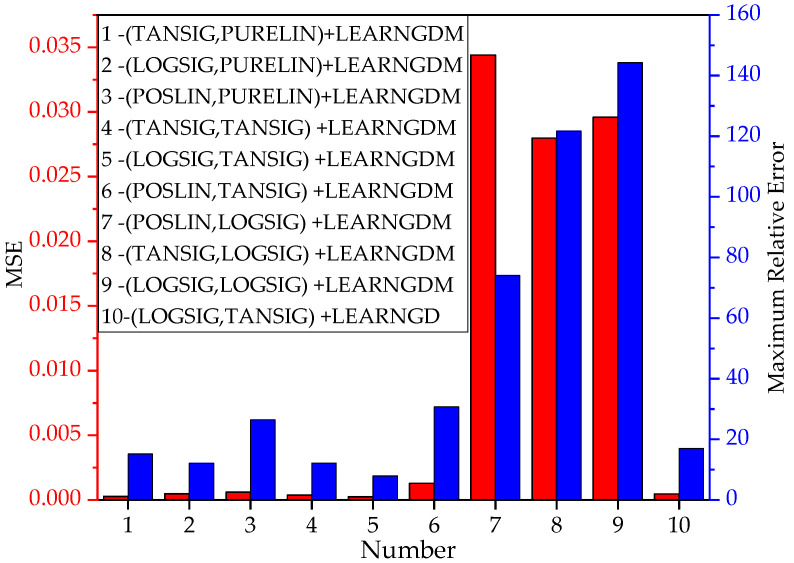
Influence of function combinations on training results.

**Figure 5 materials-15-04491-f005:**
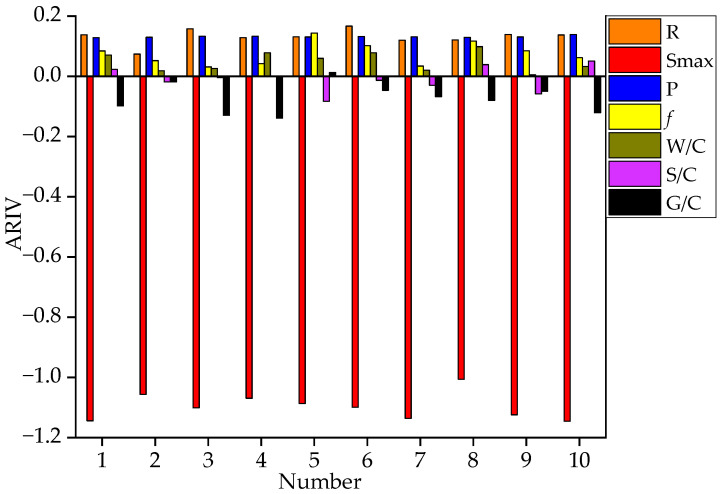
ARIV under 10 random training networks.

**Figure 6 materials-15-04491-f006:**
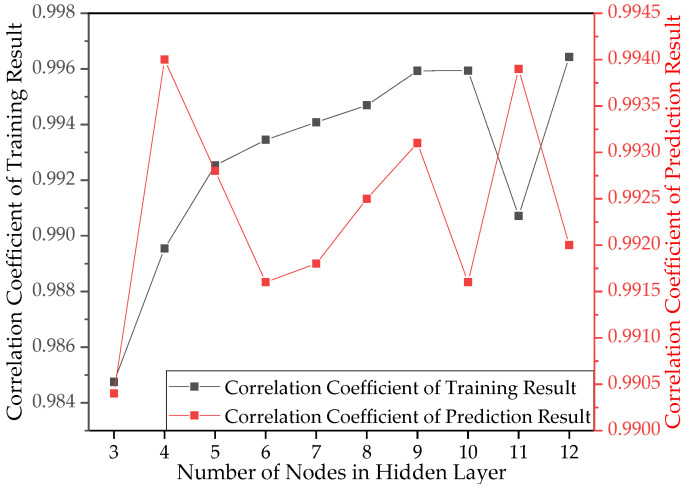
Correlation coefficients under different network structures.

**Figure 7 materials-15-04491-f007:**
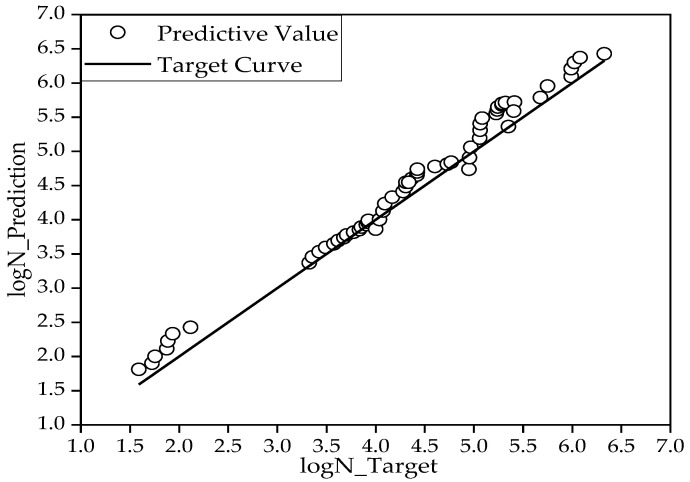
Generalization effect of BR-BPNN.

**Figure 8 materials-15-04491-f008:**
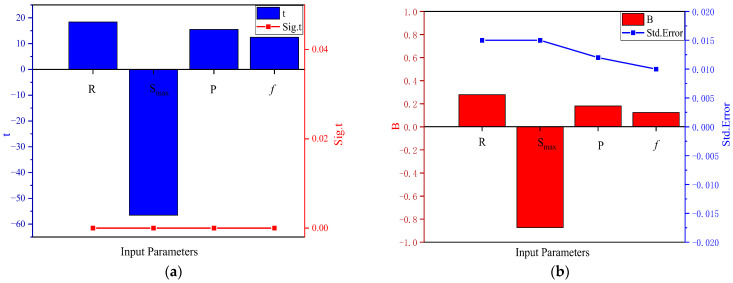
Analysis results from T test: (**a**) T statistic and its significance; (**b**) Partial regression coefficient and its Std.error.

**Figure 9 materials-15-04491-f009:**
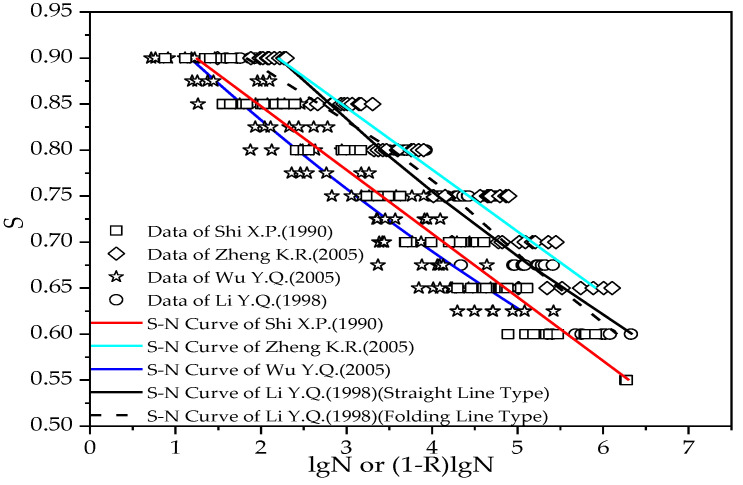
Test data and S-N curve from literature.

**Figure 10 materials-15-04491-f010:**
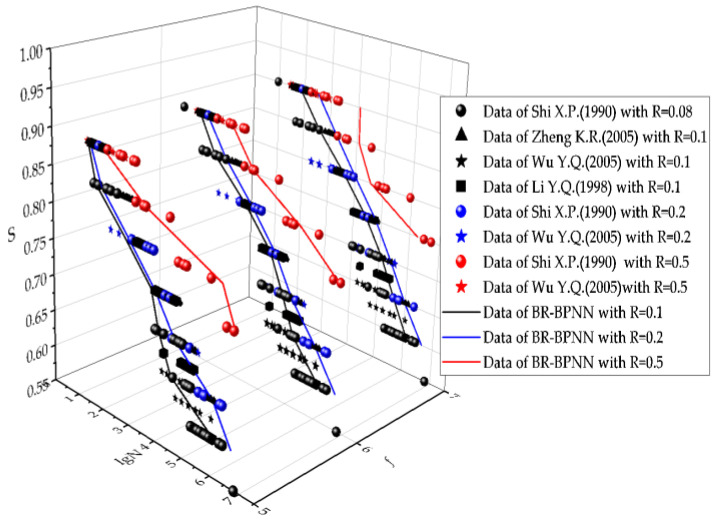
S-N curve under 50% guarantee rate predicted by BR-BPNN.

**Figure 11 materials-15-04491-f011:**
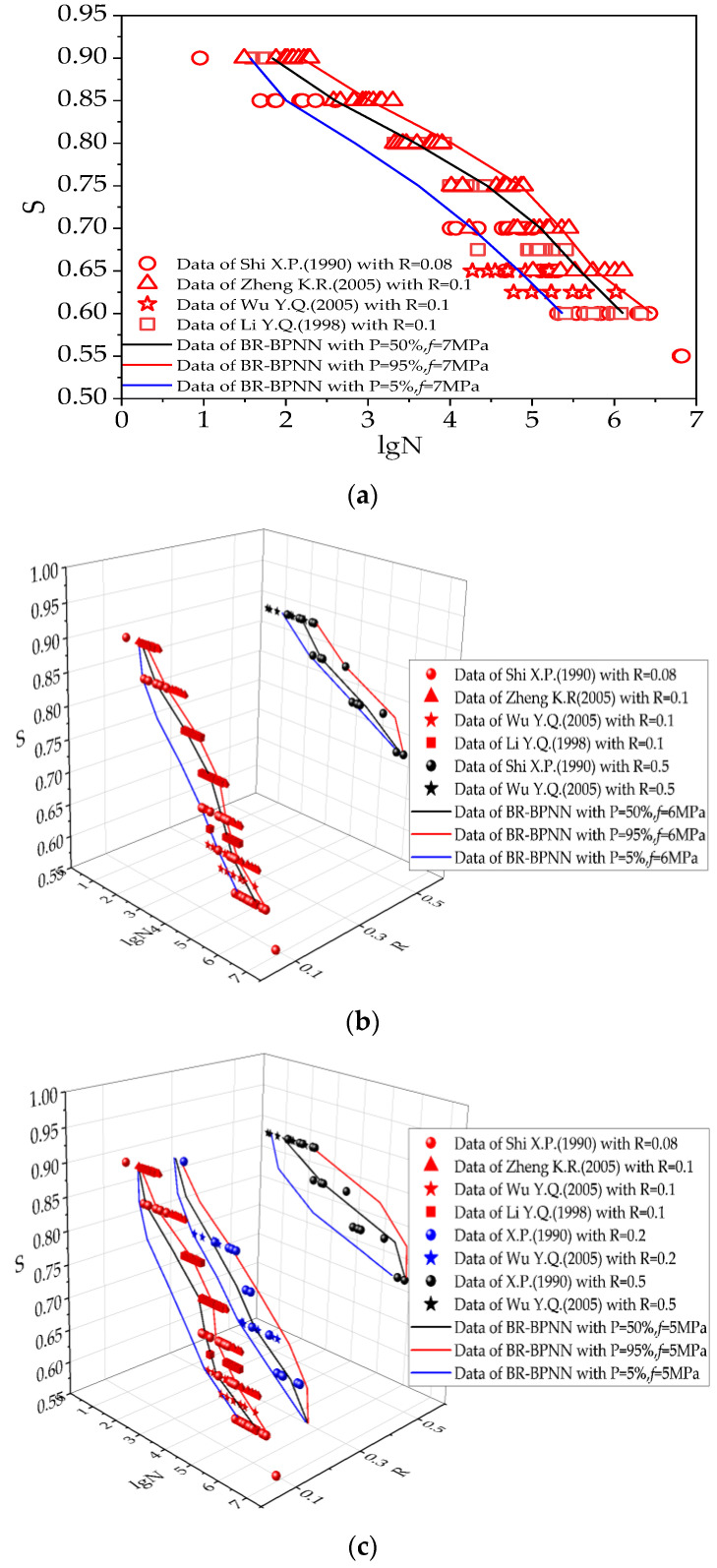
Probabilistic S-N curves predicted by BR-BPNN: (**a**) Probability analysis for *f* = 7 MPa; (**b**) Probability analysis for *f* = 6 MPa; (**c**) Probability analysis for *f* = 5 MPa.

**Figure 12 materials-15-04491-f012:**
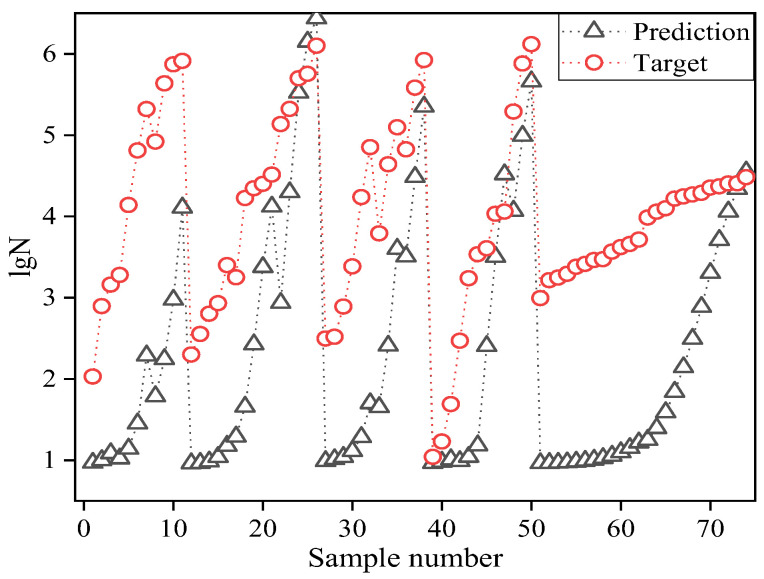
Prediction of axial tensile fatigue based on the 4-input parameters flexural fatigue network.

**Figure 13 materials-15-04491-f013:**
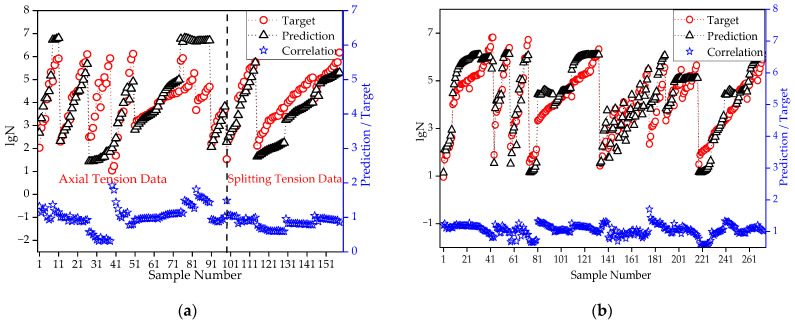
Mutual prediction of flexural and tensile fatigue life using BR-BPNN: (**a**) Prediction of tensile fatigue based on flexural fatigue network; (**b**) Prediction of flexural fatigue based on tensile fatigue network.

**Figure 14 materials-15-04491-f014:**
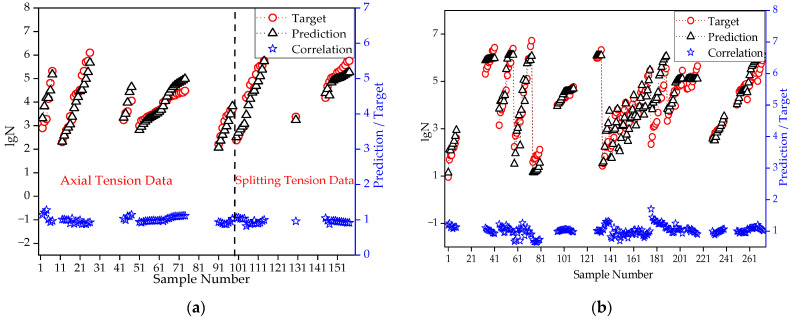
Mutual prediction of flexural and tensile fatigue life using BR-BPNN with data from Reference [43] excluded from analysis: (**a**) Prediction of tensile fatigue based on flexural fatigue network; (**b**) Prediction of flexural fatigue based on tensile fatigue network.

**Table 1 materials-15-04491-t001:** Flexural fatigue experimental data.

Purpose	Reference	Smax	R	*f* (MPa)	W/C	S/C	G/C	Equations of S-N Curves
Network accuracy	Shi et al. [32]	0.65~0.9	0.08, 0.2, 0.5	6.08	0.45	1.18	2.74	(1) lgS = 0.01069−0.04093(1−R)lgN (2) S = 0.9860−0.06919 (1−R) lgN
	Zheng [33]	0.65~0.9	0.1	7.65	0.35	1.47	2.40	(3) S = 1.04808−0.06731lgN
	Wu et al. [34]	0.625~0.9	0.1~0.5	5.1	0.45	1.40	3.27	(4) lgS = 0.002−0.0408(1−R)lgN
Generalization capability	Li et al. [35]	0.6~0.9	0.1	7.68	0.40	1.16	2.47	(5) lgS = 0.0483−0.0426lgN $(6)\begin{array}{l} \mathrm{lgS}\;=\;0.0089-0.0299\mathrm{lgN}\;S\geq 0.78\\ \mathrm{lgS}\;=\;0.0888-0.0504\mathrm{lgN}\;S\leq 0.78 \end{array}$

**Table 2 materials-15-04491-t002:** Tensile fatigue experimental data.

Loading Type	Reference	Smin	Smax	*f* (MPa)	W/C	S/C	G/C
Splitting tension	Lu et al. [38]	0.15	0.7~0.85	2.63	0.504	1.731	3.013
	Yun K.K. [39]	0.07 0.08 0.09	0.7 0.8 0.9	4.1	0.423	2.005	3.506
Axial tension	Song et al. [40]	0, 0.15, 0.3	0.65~0.85	2.45	0.504	1.731	3.013
	Meng [41]	0.22, 0.27	0.75~0.85	2.69	0.504	1.731	3.013
	Wang [42]	0.1	0.7~0.9	3.06	0.360	1.403	2.494
	Huang, L.X. [43]	0	0.3~0.6	2.01	0.410	2.005	3.506

**Table 3 materials-15-04491-t003:** Selection of BR-BPNN functions.

Activation Function		Learning Function	Training Function	Performance Function
Hidden Layer	Output Layer			
LOGSIG	TANSIG	LEARNGDM	TRAINBR	MSE

**Table 4 materials-15-04491-t004:** ARIV analysis of input parameters for the final BR-BPNN.

Input Parameter	Analysis Result of ARIV
R	0.143
Smax	−0.871
P	0.122
*f*	0.116

**Table 5 materials-15-04491-t005:** Connection weights of the BR-BPNN.

Weight		Neuron 1	Neuron 2	Neuron 3	Neuron 4	Neuron 5	Neuron 6	Neuron 7	Neuron 8	Neuron 9
w^ih^	R	−2.466	0.241	−2.254	3.662	−2.485	−1.207	2.636	0.331	0.533
	S_max_	−2.447	2.019	−3.015	−5.607	1.485	1.878	0.187	2.564	−3.537
	P	0.459	1.055	−0.278	0.973	−0.903	0.818	2.046	−0.280	0.818
	*f*	0.288	−1.000	−0.041	−1.023	0.272	0.263	−1.974	0.537	0.817
w^ho^		−2.866	−1.634	−2.143	3.628	2.243	1.968	−0.822	−3.570	3.853

**Table 6 materials-15-04491-t006:** Summary of analysis results from MLR models.

Model	Dependent Variables	Independent Variables	Removed Variables	Adjusted R2	SEE	F Change	Sig. F Change
1	lgN	Smax, R, P, *f* S/C, G/C	W/C	0.937	0.05371	673.596	0.000
2		Smax, R, P, *f*	/	0.931	0.05599	923.723	0.000
3		Smax, R, W/C, S/C, G/C	/	0.875	0.442	383.921	0.000
4		W/C, S/C, G/C	/	0.047	0.20832	5.453	0.001

**Table 7 materials-15-04491-t007:** Regression coefficient analysis of Model 1.

Model	Independent Variables	Std.error	t	Sig.t	Collinear Analysis	
					Tol	VIF
2	R	0.015	19.235	0.000	0.485	2.062
	Smax	0.015	−57.089	0.000	0.634	1.577
	P	0.011	16.150	0.000	1.000	1.000
	*f*	0.030	−0.419	0.676	0.063	15.910
	S/C	0.009	3.018	0.003	0.769	1.301
	G/C	0.032	−4.876	0.000	0.062	16.108

## Data Availability

Details of flexural fatigue experimental data (274 data points), tensile fatigue experimental data (158 data points), and the four-parameter BR-BPNN model presented in this study are available upon reasonable request to the corresponding author.

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
