# Peer review of "Fatigue Factor Assessment and Life Prediction of Concrete Based on Bayesian Regularized BP Neural Network"

_materials, 2022, doi:10.3390/ma15134491_

Round 1

Reviewer 1 Report

The authors present the details of the development and implementation of a neural network used for predicting the fatigue life of concrete.  The writing is clear and the organization is done well.  There are a few items that need to be addressed:

1) In figures of fatigue data, magenta and red are too close to each other to be distinguishable, particularly when some of the plots are so small.  Select different colors and, preferably, make sure that the symbols associated with each color are also different from each other in case a reader does not view the paper in color.

2) My single most significant concern with respect to the data presented is with the parameter P.  Are the authors certain this is not actually survival probability, not failure probability?  It seems extremely odd that failure probability would be positively correlated with fatigue life (i.e., a sample with a higher probability of failure would somehow last longer under fatigue loading).

3) Failure strength of concrete (symbol f used by the authors) is strongly correlated to parameters like water-cement ratio (W/C), etc.  Surprisingly, the authors' analysis found W/C to be a 'noise' parameter.  Looking at Table 6, Smax and R can be independently defined during fatigue testing.  However, P and f are dependent on both the applied loading and the material properties.  Model 1 thus includes both loading and material properties.  Similarly, Model 3 includes both loading and materials properties.  However, Model 2 included only material properties.  It is not surprising, then, that Model 2 would perform less well than the others.  Was a model tested that included Smax, R, W/C, S/C, and G/C only?  That, I think, would be a better test of the value of the material-only parameters because you have removed P and f which are a combination of both load and material considerations.  I don't think the data presented actually support the conclusion that W/C is 'noise'.

Reviewer 2 Report

Considering fatigue life of concrete is affected by loading and material properties that are nonlinearly interwoven, this paper utilizes BR-BPNN to evaluate parameters affecting fatigue life and predict fatigue life under different experimental conditions. The paper is interesting and could be published in the Journal of materials in the reviewer's view if the following comments and other reviewers, comments to be fulfilled.

- Abstract could be more informative by providing results. I prefer to see some results in the abstract.

- Abstract needs to be clarified in a sense to differentiate this study from other published work in this field. The objectives of this work are evidently given in abstract but the sentence could be modified to give more clarity.

- The abstract is looking weak owing to the unavailability of background knowledge of this study. Add comprehensive line at the start of the abstract about the background history of the work. Also, add some key values from results and highlight the novelty of this work clearly. The ending of this section is quite abrupt. Complete the abstract with a conclusive on this work and its findings.

- Please use some innovative keywords.

- Please mention your study limits in the abstract.

- The introduction needs to be more emphasized on the research work with a detailed explanation of the whole process considering past, present and future scope. How the present study gives more accurate results than previous studies? It needs to be strengthened in terms of recent research in this area with possible research gaps. It is strongly recommended to add a recent literature.

- Selected references are quite old, which from the one point of view is good, since the authors cited necessary references to define a research problem, while from the other hand, lack of recent references may indicate an insufficiently performed literature review. Try to refer to some recent and up-to-date research papers related to the topic, especially in recent years like those related to fatigue:

- Fatigue energy dissipation and failure analysis of angle shear connectors embedded in high strength concrete

- Fatigue energy dissipation and failure analysis of channel shear connector embedded in the lightweight aggregate concrete in composite bridge girders

Please also read the following papers about neural networks:

- Assessment of longstanding effects of fly ash and silica fume on the compressive strength of concrete using extreme learning machine and artificial neural network

- Prediction of concrete strength in presence of furnace slag and fly ash using Hybrid ANN-GA (Artificial Neural Network-Genetic Algorithm)

- Application of a hybrid artificial neural network-particle swarm optimization (ANN-PSO) model in behavior prediction of channel shear connectors embedded in normal and high …

- A novel approach to predict shear strength of tilted angle connectors using artificial intelligence techniques

- Assessment of longstanding effects of fly ash and silica fume on the compressive strength of concrete using extreme learning machine and artificial neural network

- Prediction of concrete strength in presence of furnace slag and fly ash using Hybrid ANN-GA (Artificial Neural Network-Genetic Algorithm)

- The authors have to explain what is the new here in comparison with the previous studies.

- The novelty of the current work should be highlighted in the introduction

- Please try to mention a problem that needs solving - in other words, the research question underlying your study more clear.

- Please kindly make revision on the language of the paper presentation. There are still some minor typos and grammatical errors.

- Please improve the quality of pictures in the manuscript.

Please explain your methodology in more detail and your mention why did you choose this method?

- Try to refer to the papers used this methodology.

- Please report enough detail in order to replicate the study.

- Plots of all Figures need to be uniformed in size and style

- Equation 3: what is λ?

- Can the correlation between output and input significantly affect the training quality?

- Can you explain why the number of specimen in literature are limited?

- Authors are advised to give a reference for this statement “Learning rate affects step adjustment of the loss function.”

- Plots of all Figures need to be uniformed in size and style

- The unit of each parameter needs to be added for each equation, even if dimensionless

- Can you please explain figure 12 in more detail?

- How can we increase training accuracy in your study?

- Please compare your results with some papers in the related field and explain the similarity and differences between your results and theirs.

- The discussion of the comparison results may be strongly extended, by providing proper considerations to each plotted graph

- Please improve the quality of Figures, some of them are blur.

- Please compare your results with some related and up-to-date papers, like:

- Ductility and strength assessment of HSC beams with varying of tensile reinforcement ratios

- An experimental study on the failure modes of high strength concrete beams with particular references to variation of the tensile reinforcement ratio

- Effect of pumice powder and nano-clay on the strength and permeability of fiber-reinforced pervious concrete incorporating recycled concrete aggregate

- Analyzing shear strength of steel-concrete composite beam with angle connectors at elevated temperature using finite element method

- Corrosion resistance evaluation of rebars with various primers and coatings in concrete modified with different additives

- Evaluating the use of recycled concrete aggregate and pozzolanic additives in fiber-reinforced pervious concrete with industrial and recycled fibers

- Effect of progressive shear punch of a foundation on a reinforced concrete building behavior

- Monotonic behavior of C and L shaped angle shear connectors within steel-concrete composite beams: an experimental investigation

- Application of waste tire rubber aggregate in porous concrete

- Potential of soft computing approach for evaluating the factors affecting the capacity of steel–concrete composite beam

Major Comment: The conclusion should be an objective summary of the most important findings in response to the specific research question or hypothesis. A good conclusion states the principle topic, key arguments and counterpoint, and might suggest future research. It is important to understand the methodological robustness of your study design and report your findings accordingly. Please improve your conclusion section.

- Please mention your study limits and suggest some future research topics.

- The authors are advised to write the conclusion in a comprehensive way, it should contain key values, suitability of the applied method, contributions and possible future work. 

- The discussion of the comparison results Section may be strongly extended, by providing proper considerations to each plotted graph

- The "Conclusions" may be provided as a "uniform text", rather than using bullet points. The weaknesses of the work and the future improvements should be added in this section

- The authors are advised to revise references, including the latest references. Please see some suggestions in the comments for the ‘introduction’ section like:

- Computational Lagrangian Multiplier Method by using for optimization and sensitivity analysis of rectangular reinforced concrete beams

- Shear capacity equation for channel shear connectors in steel-concrete composite beams

- Investigation on composite polymer and silica fume-rubber aggregate pervious concrete

- Numerical analysis of tilted angle shear connectors in steel-concrete composite systems

- Comparative performance of channel and angle shear connectors in high strength concrete composites: An experimental study

- Investigation of through beam connection to concrete filled circular steel tube (CFCST) column

- Potential of adaptive neuro fuzzy inference system for evaluating the factors affecting steel-concrete composite beam's shear strength

- An evolutionary fuzzy modelling approach and comparison of different methods for shear strength prediction of high-strength concrete beams without stirrups

- Efforts should also be applied to improve English writing. This paper is not written to a high level. The text seems to be too 

Reviewer 3 Report

there are novelty in this research.

please describe table 2 in the text.

Table 2. Tensile fatigue experimental data.  must be update. the data is derived frope 4 paper that published in 2002 to 2010. please use recent research.

aggregate mixture must be describe similar to folowing research:

https://trid.trb.org/view/1657991

please use similar paper in research history section.
